# Pure Rotational Spectrum of Benzophenone Detected by Broadband Microwave Spectrometer in the 2–8 GHz Range

**Haoyang Tan, Miaoling Yang, Chenbo Huang, Shengwen Duan, Ming Sun \*, Qian Chen \*, Chao Jiao \* and Yi Wu**

School of Electronic and Optical Engineering, Nanjing University of Science and Technology, Nanjing 210094, China; tanhaoyang@njust.edu.cn (H.T.); yangmiaoling@njust.edu.cn (M.Y.); chaoshuang@njust.edu.cn (C.H.); 117104021762@njust.edu.cn (S.D.); 118104021806@njust.edu.cn (Y.W.)
**\*** Correspondence: msun@njust.edu.cn (M.S.); chenq@njust.edu.cn (Q.C.); cjiao@njust.edu.cn (C.J.)

**Abstract:** The investigation on microwave spectrum of benzophenone was conducted with a recently constructed broadband chirped-pulse Fourier transform microwave spectrometer with a heating nozzle in the 2–8 GHz range. In this work, 138 b-type pure rotational transitions were assigned to bridge the measuring gap in the microwave region. The rotational constants for benzophenone were accurately determined by a combined microwave data fitting with frequency coverage between 2–14 GHz and have the following values: $A = 1692.8892190(119)$ MHz, $B = 412.6446602(43)$ MHz and $C = 353.8745644(43)$ MHz.

**Keywords:** benzophenone; rotational spectroscopy; microwave; cp-FTMW

---

## 1. Introduction

Benzophenone is an aryl ketone that could be prepared by a Friedel–Crafts acylation of benzoyl chloride with benzene [1] or by the oxidation of diphenylmethane [2]. Benzophenone is of great use in industrial applications to manufacture agricultural chemicals such as insecticides, and pharmaceuticals such as hypnotics [3]. Due to its photochemical reactivity, benzophenone can work as a photoinitiator and an ultraviolet curing agent [3].

Benzophenone might exist in galaxies similar to polycyclic aromatic hydrocarbons (PAHs), chemicals believed to be widespread through circumstellar envelopes and interstellar medium (ISM) of carbon rich stars [4,5]. Different from many PAHs with tiny dipole moments, benzophenone with a dipole moment about of 3.0 Debye has decent signals of rotational transitions for molecular hunting in deep space by radio telescopes. Benzophenone could also occur in the environment, such as ambient atmosphere, to bring health risks [6,7]. Under these conditions, laboratory rotational spectroscopic data on benzophenone can be a useful guide for radioastronomical and atmospheric searches for this molecule.

Currently, benzophenone has been investigated by microwave/millimeter-wave spectroscopy in the 8–14 GHz and 60–73 GHz region [8,9]. The purpose of this work is to extend the laboratory measurements of the rotational spectrum of benzophenone at lower frequency to further support the astrophysical observation and atmospheric remote sensing of benzophenone molecule. The measurement and analysis of the detected spectra of benzophenone in the vibrational ground state between 2 and 8 GHz are reported.

## 2. Experiment

### 2.1. Experimental Instrument and Methods

The microwave spectrum of benzophenone in the 2–8 GHz range was collected by a recently developed broadband chirped-pulse Fourier transform microwave (cp-FTMW) spectrometer with a heating nozzle at Nanjing University of Science and Technology (NJUST) as shown in Figure 1, which has been described in detail previously [10,11]. Basically, a vacuum sample chamber (1.0 m in length and 0.5 m in diameter) housing a reflection spherical aluminum mirror (350 mm in diameter and 800 mm in radius of curvature), a sample pulsed solenoid nozzle ((Parker Series 9, 1.0 mm in diameter)), and a feedhorn antenna makes the main mechanical part of the spectrometer. The nozzle, penetrating the center of the mirror, is in a coaxial arrangement with the mirror, the antenna, and the chamber. The vacuum sample chamber was pumped by a molecular pump to maintain a background pressure about $1 \times 10^{-5}$ Pa.

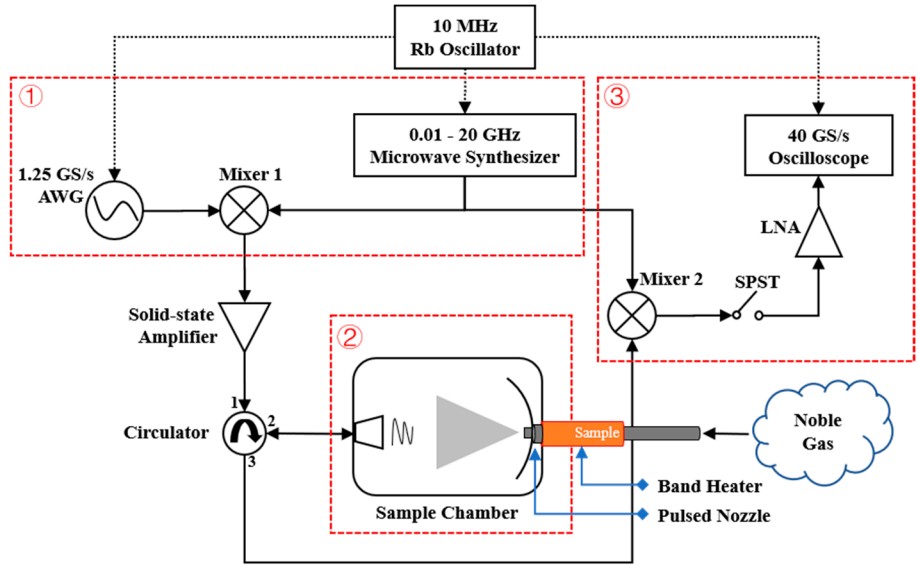

**Figure 1.** The schematic diagram of the cp-FTMW spectrometer, which is divided into three parts: (1) microwave signal source; (2) sample vacuum chamber; (3) signal detection system.

The spectrometer is based on the linear frequency sweeps (chirped pulses) generated by an arbitrary wave generator (AWG), as described by Pate and others [12,13], but equipped with an economic quadrature detection plan. In this work, a homodyne circuit design was applied to the cp-FTMW spectrometer system: the continuous microwave generated by the microwave source was split in two through a power distributor. Half of the microwave up-converted the broadband source generated by arbitrary waveform generator to excite the sample, and the other half down-converted the molecular excited signal to the oscilloscope acceptable bandwidth range. In this work, a 5 μs, 0–500 MHz chirped pulse from AWG (Siglent SDG6052x-e, 1.25 GS/s) was mixed in mixer 1 (Marki M1-0220LA Mixer) with the LO output from a microwave synthesizer (Anapico Apsyn420, 0.01–20 GHz) to generate an exciting pulse with 1 GHz chirp bandwidth, that was then amplified using a solid-state amplifier (BONN, 5 W) before being coupled into a custom-built vacuum sample chamber containing one high gain horn antenna and a reflection spherical aluminum mirror for broadcasting and receiving the molecular emission signals. The molecular signal, i.e., the free induction decay (FID), was first amplified by a low-noise amplifier (Miteq AFS44 LNA, 1–18 GHz) before being split by a homemade quadrature detection structure to achieve image rejection. The image rejected FID signals were further amplified by two identical low-noise amplifiers ((Mini-Circuits ZFL 500LN+, 0.1–500 MHz), which were subsequently sampled by the two channels of a digitizer (Spectrum M4i.2223, 2.5 GS/s,

1.5 GHz bandwidth). Fast Fourier transformation of the resulting free induction decay leads to the broadband spectrum over the full bandwidth of the chirp. The whole sequence was repeated to produce multiple FIDs so that the resulting spectrum gave an improved signal-to-noise ratio. Furthermore, the DF4351 frequency synthesizer, the SDG6052x-e AWG and the M4i.2223 digitizer were referenced to a 10 MHz rubidium standard oscillator (SRS, FS725) for external stability. In order to avoid damage to the downstream amplifier and oscilloscope caused by high-power microwave pulses, a single-pole single-throw electronic switch (SPST, F9114A, 1–18 GHz) was set behind mixer 2 (Marki M1-0220LA Mixer), which could be closed in time to protect the downstream electronic equipment.

Benzophenone (99.9% purity) was purchased from Aldrich without further purification. A sample gas mixture was obtained by flowing high purity argon (99.9% purity) through benzophenone right before the Parker nozzle, which was coupled with a band heater to obtain sufficient vapor pressure for solid chemicals. The temperature of the heater was regulated by a temperature controller to maintain a set temperature up to 250 °C. In this work, the temperature of the heating nozzle was brought up to 150 °C to show a decent molecular signal [8], with a pressure of 0.6 MPa of argon allowed to flow through the heated sample as a carrier gas for the supersonic expansion. The molecular excited signal FID was collected at a sampling rate of 2.5 GS/s for 10 μs, and the spectral resolution could reach about 150 kHz. We repeated the data acquisition sequence and averaged a total of 10,000 shots (230,000 FIDs) to produce better signal-to-noise in the spectrum.

In this paper, our experiment was carried out in the 2–8 GHz range of the benzophenone molecule to test our quadrature detection scheme and to verify the molecule's spectroscopic constants as well. Altogether, ten scans of 1 GHz range each were used to actually cover the 2–12 GHz region, and the whole spectrum is shown in Figure 2. One such 1 GHz experimental spectrum is displayed in Figure 3, with a portion presenting assigned rotational transitions in Figure 4.

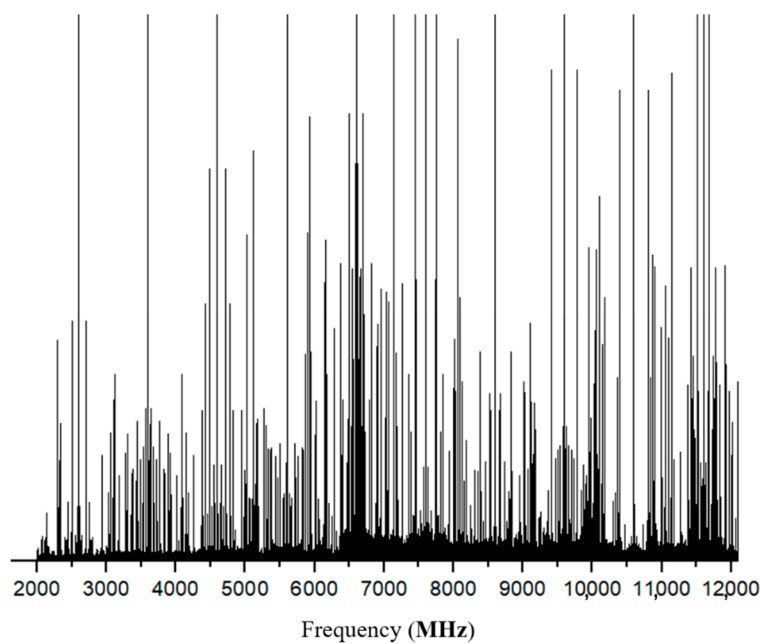

**Figure 2.** Experimental spectrum of benzophenone in the 2–12 GHz range.

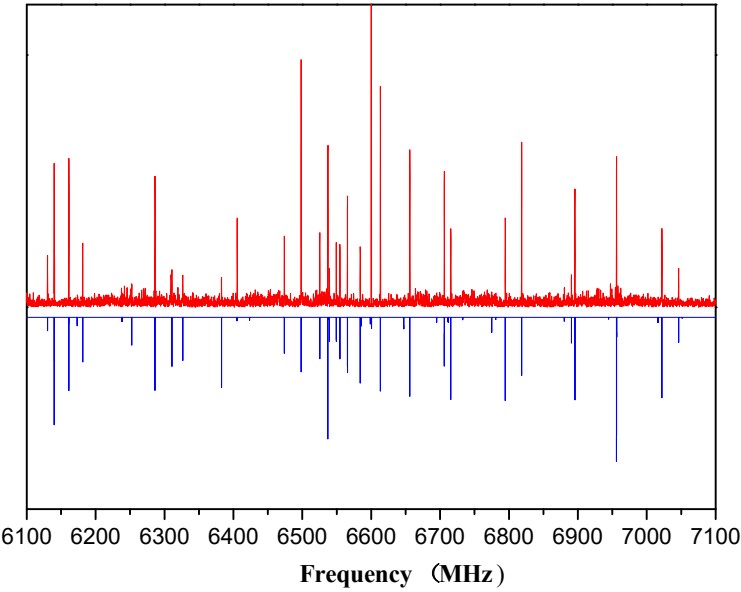

**Figure 3.** Experimental spectrum of 10,000 acquisition cycles of benzophenone with 1 GHz bandwidth (top, red, LO = 6.6 GHz), compared to a simulation at 10 K generated using the constants from [8] (bottom, blue).

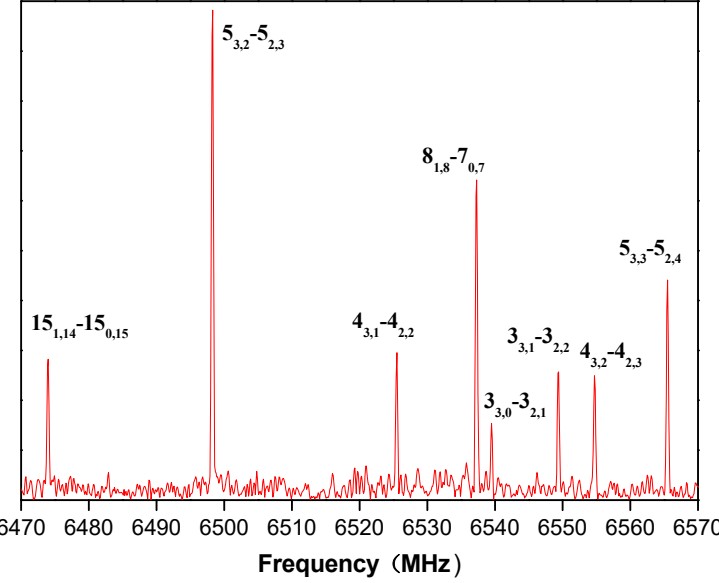

**Figure 4.** A partial experimental spectrum of benzophenone with 8 assigned rotational transitions.

## 2.2. Theoretical Calculation

Based on ab initio [14] and density-functional theory (DFT) calculations [15,16], the isolated structure of benzophenone was optimized by Gaussian09 software package [17]. The ab initio calculation was carried out using the second order Møller–Plesset perturbation theory (MP2) [14] while the DFT calculation was performed using the Becke, 3-parameter, Lee–Yang–Parr (B3LYP) [15,16]. In both cases, the cc-pVDZ basis set [18] was employed for calculation. Thanks to these calculations, we ascertained the molecular constants or rotational constants required for guiding the initial spectroscopic assignment and fitting procedures. A three-dimensional structure diagram of global minimum geometries for benzophenone on the optimized MP2/cc-pVDZ theory level is shown in Figure 5, which is consistent with the expected paddle-wheel type like $C_2$ structure. Table 1 shows the rotational parameters obtained at the MP2/cc-pVDZ and B3LYP/cc-pVDZ levels of theory, as well as molecular parameters of benzophenone predicted by

PMIFST (Principal Moments of Inertia from Structure) [19]. The three-dimensional coordinate matrix calculated by Gaussian09 based on MP2/cc-pVDZ and B3LYP/cc-pVDZ levels of theory are supplied in the Supplementary Materials.

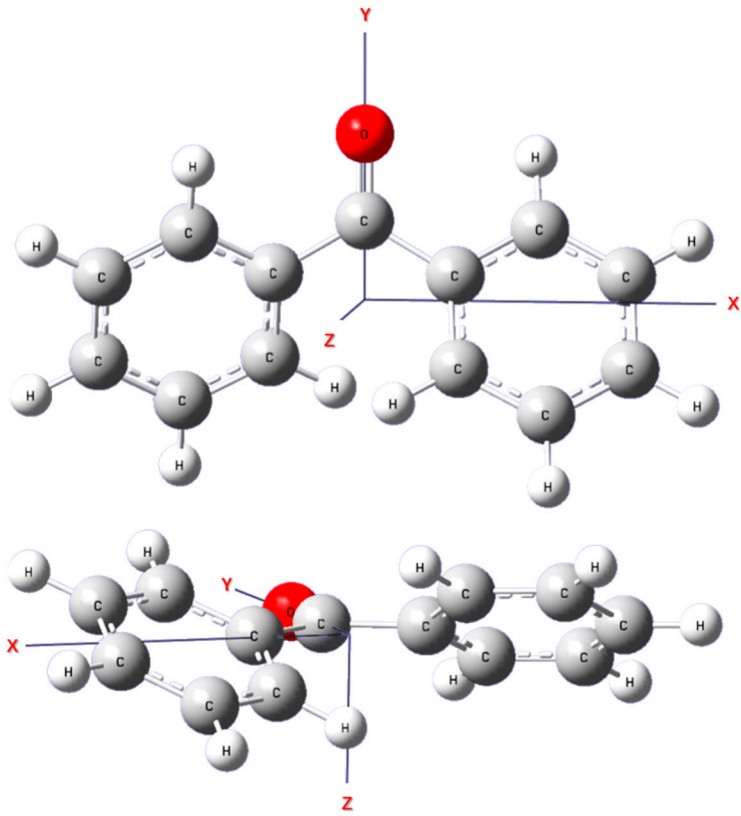

**Figure 5.** Optimized global minimum geometries for benzophenone on the MP2/cc-pVDZ theory level.

**Table 1.** The computational rotational constant, moment of inertia, planar moment of inertia, dipole moment and asymmetric parameter of benzophenone.

| Parameters | MP2/cc-pVDZ | B3LYP/cc-pVDZ |
|:---:|:---:|:---:|
| A(MHz) | 1666.94 | 1716.36 |
| B(MHz) | 410.30 | 405.16 |
| C(MHz) | 349.82 | 345.92 |
| I. a(amu·Å$^2$) | 303.18 | 294.45 |
| I. b(amu·Å$^2$) | 1231.72 | 1247.36 |
| I. c(amu·Å$^2$) | 1444.68 | 1460.98 |
| P. a(amu·Å$^2$) | 1186.61 | 1206.95 |
| P. b(amu·Å$^2$) | 258.07 | 254.03 |
| P. c(amu·Å$^2$) | 45.11 | 40.42 |
| $\mu_b$(Debye) | 3.20 | 2.82 |
| $\Delta$(amu·Å$^2$) [a] | −90.22 | −80.83 |
| kappa | −0.91 | −0.91 |

[a] $\Delta = I_c - I_a - I_b$.

## 3. Assignment and Results

The analysis of the pure rotational spectra in the microwave region was carried out using Watson S-reduced effective Hamiltonian [20]. Entire observed transitions were fitted and assigned using the $I^r$ representation in the Pickett SPFIT/SPCAT (spectral fitting analysis software) [21]. The final fitting results were reformatted to convert SPFIT errors to standard errors [22]. All of the computational and

experimental studies have proven that benzophenone in the ground state is of $C_2$ symmetry, but with the two phenyl groups in a paddle-wheel-like orientation [8,9,23,24]. In that case, the permanent electric dipole moment of benzophenone lies along the *b*-principle axis allowing transitions obeying *b*-type selection rules (*eo* ↔ *oe* and *oe* ↔ *eo*, *e* for even *o* for odd values of the quantum numbers $K_a$ and $K_c$) [25]. The assignment of the frequency transition lines in the 2–8 GHz range was assisted by the rotational constants fitted from the previous microwave and mm-wave experimental data, as shown in Figure 3. The top spectrum with 1 GHz bandwidth showed a strong leaking LO signal at the center, 6.6 GHz, with a portion displaying assigned rotational transitions in Figure 4. The bottom trace in Figure 3 simulated by using the parameters from the van Wijngaarden group [8] agreed well with our experimental data, especially in frequency. A total of 138 new *b*-type transitions were assigned in the 2–8 GHz range with $J_{max} = 23$ and $K_{a\ max} = 7$. The newly assigned transitions, along with the microwave data measured by the van Wijngaarden group between 8 and 14 GHz [8], were fitted and the obtained spectroscopic constants are given in Table 2. The root-mean-square (RMS) deviation $\sigma$ of the combined fit is 2.6 kHz, with the typical linewidths of about 150 kHz. All the newly assigned rotational transitions of benzophenone in 2–8 GHz are provided in Supplementary Materials.

**Table 2.** Fit spectroscopic parameters for benzophenone.

| Parameters | Combined | Previous Work [a] | |
|---|---|---|---|
| Frequency range (GHz) | 2.0–14.0 | 8.0–14.0 | 60.0–73.0 |
| A (MHz) | 1692.889219(31) | 1692.88929(23) | 1692.916(35) |
| B (MHz) | 412.644660(11) | 412.644594(92) | 412.620(17) |
| C (MHz) | 353.874564(11) | 353.874501(88) | 353.884(17) |
| $D_J$ (kHz) | 0.010635(19) | 0.01053(14) | 0.0122(17) |
| $D_{JK}$ (kHz) | −0.043444(83) | −0.04372(54) | −0.0517(60) |
| $D_K$ (kHz) | 0.41417(59) | 0.4195(43) | 0.462(62) |
| $d_1$ (kHz) | −0.0014406(23) | −0.001442(17) | |
| $d_2$ (kHz) | −0.00099(15) | −0.0003385(68) | |
| $\sigma$(kHz) | 2.6 | 8 | 110 |
| N | 304 | 166 | 54 |

[a] Constants are from Ref. [8,9].

## 4. Discussion and Analysis

Pure rotational transitions of the ground state benzophenone were accurately measured by the cp-FTMW with a quadrature detection design in this work. The new assignment between 2 to 8 GHz bridged the measuring gap in the lower microwave range for this molecule. A combined fit covering 2–14 GHz allowed for more precise determination of the rotational constants. Comparing the rotational constants with that from higher microwave frequency data fitting [8,9], the accuracy of our combined fitting is significantly improved, as shown in Table 2. The RMS deviations of the fit of 2.6 kHz is less than two percent of the ~150 kHz experimental linewidths, revealing that the effective Hamiltonian employed here has accurately described the observed spectral features in the microwave range. However, for certain centrifugal distortion constants, such as $d_2$, a global fitting from higher microwave frequency data [8,9] can definitely provide more accurate results due to the high $K_a$ and $K_c$ transitions available.

Including this work, plenty of computational investigations have been performed on the benzophenone [8,9,23,24]. In summary, ab initio and Density Functional Theory methods were mostly employed to optimize its structure, undoubtedly providing the paddle-wheel-like $C_2$ geometry for a global minimum with a single component of the dipole moment along the b principle axis. For the hindered internal rotation of the two phenyl groups, the latest MP2/cc-pVQZ structure from the van Wijngaarden group [8] by the second moment calculation determined a 32.9° torsional angle of the phenyl groups out of the a, b-principle plane in the gas-phase. In addition, two higher energy

conformers regarding such hindered rotation with one T-like $C_s$ symmetry and one $C_{2v}$ symmetry were predicted to stay 1084 cm$^{-1}$ and 4267 cm$^{-1}$ above the global minimum, respectively, which excluded resolvable tunneling splitting associated with the internal rotations in the microwave spectrum, and was confirmed by the free-jet experiments from our group and others [8,9].

## 5. Conclusions

The current study presented a pure rotational investigation on benzophenone molecules in the lowest microwave frequency region so far. The newly built cp-FTMW spectrometer with a quadrature detection scheme and a heating nozzle demonstrated its capability to measure such non-volatile molecules in a supersonic expansion. Our accurately determined rotational constants can provide the basis for the theoretical calculation and rovibrational studies of torsional modes. The results from this work also have the potential to guide the atmospheric detection and deep space search for this molecule.

**Supplementary Materials:** The following are available online at http://www.mdpi.com/2076-3417/10/23/8471/s1, Table S1: X, Y, Z coordinates of the optimized structure using MP2/cc-pVDZ method; Table S2: X, Y, Z coordinates of the optimized structure using B3LYP/cc-pVDZ method; Table S3: Observed Benzophenone transitions.

**Author Contributions:** H.T., M.Y., C.H., S.D., C.J., Y.W. designed and performed the experiments. H.T., M.Y., C.H., S.D. analyzed the data and wrote the article. M.S., Q.C. and C.J. reviewed and edited the manuscript. C.J. responded to the reviews and revised the manuscript. All authors have read and agreed to the published version of the manuscript.

**Funding:** This research was funded by the National Natural Science Foundation of China (Nos. 61627802, U1531107), the National Undergraduate Training Program for Innovation and Entrepreneurship (201910288031Z).

**Acknowledgments:** The authors would like to thank the editors and the reviewers for their comments on an earlier draft of this article.

**Conflicts of Interest:** The authors declare no conflict of interest.

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
