# Peer review of "Pure Rotational Spectrum of Benzophenone Detected by Broadband Microwave Spectrometer in the 2–8 GHz Range"

_applsci, doi:10.3390/app10238471_

Round 1
Reviewer 1 Report
This paper entitled:”Pure Rotational spectrum of Benzophenone detected by broadband microwave spectrometer in the Range of 2-8 GHz” by Haoyang Tan, Miaoling Yang, Chenbo Huang, Shengwen Duan, Ming Sun, Qian Chen, Chao Jiao and Yi Wu reports a combined theoretical/esperimental investigation of Benzophenone.
In particular, thanks to the use of a newly constructed broadband chirped-pulse Fourier transform microwave spectrometer, the extension of the rotational spectrum of benzophenone (already observed at higher frequency) in 2-8 GHz range has been performed, providing more accurate rotational parameters, also taking into account the previous data.
I think the paper can be suitable for publication in the journal but only after the authors have addressed these points:
- The authors write (lines 24-26):” Benzophenone might exist in galaxies like polycyclic aromatic hydrocarbons (PAHs) widespread through circumstellar envelopes and interstellar medium (ISM) of carbon rich stars [4,5]. The two reported references are not related to the astrochemical sentence the authors wrote. Please either correct them or remove the sentence.
- The authors write(lines 32-33):” Up to date, benzophenone has been investigated by microwave spectroscopy [5] and millimeter-wave spectroscopy [6].” In my opinion is better to write the frequency ranges since all the already measured and the present rotational transitions are within the microwave range.
- Please re-write the sentence (lines 33-35) as something like:” This work aims to extend the laboratory measurements of the rotational spectrum of benzophenone at lower frequency…..”
- Please rephrase (line 40):”The microwave spectrum of benzophenone in the range of 2-8 GHz…” in “ The microwave spectrum of benzophenone in the 2-8 GHz range..”
- I think it is better to rephrase the sentence:” The instrumentation of the spectrometer was based on the linear frequency sweeps (chirped pulses) generated by an arbitrary wave generator (AWG) achieved by the Pate and many other groups [9,10], but equipped with an economic quadrature detection plan.” as something like:” The spectrometer is based on the…. , as described by Pate and others [ ], “.
- I think there is a typo in “Ciculator” in Figure 1.
- What is LO output?
- Please rephrase the sentence:” A gas mixture of noble gas seeded with sample molecules was allowed to enter the vacuum chamber through a heating nozzle, i.e. a pulsed solenoid valve (Parker Series 9, 1.0 mm diameter) coupled with heating container (up to 250 ˚C) to obtain sufficient vapor pressure for seed chemicals.” as something like:” A gas mixture noble gas/sample has been obtained by flowing the….and expanded into the chamber…..”.
- Line 78/79: please correct the “lowest frequency range of benzophenone molecule” in “ the 2-8 GHz range.
- Line 80: do you mean “A total of 10 scans of 1 GHz range each” ?
- I think it is better to write in line 93:” Based on MP2 , and DFT calculations, the isolated structure of benzophenone (not phenylacetone) was optimized….”. Please note that when you optimize to a minimum, what you obtain is the equilibrium structure (no ground vibrational state).
- Please correct (line 95): “the base set cc-pVDZ [15] was applied…” in “ the cc-pVDZ basis set was employed…”.
- Please rephrase the sentence (line 95-99) as something like:” Thanks to these calculations, the rotational parameters required for guiding the….”
- In line 98: are they “planar projections”?
- I think the authors can change the sentence at line 99 as (or something like): benzophenone using the optimized MP2/cc-pVDZ structure, are..”
- Line 101-103. I think is better to rephase as:” Table 1 shows the rotational parameters obtained at the MP2/cc-pVDZ and B3LYP/cc-pVDZ levels of theory as well as the moment of inertia and the planar moment of inertia of benzophenone…”
- Please rephase the caption of Figure 5 (please don’t use opt, but explicit it). Are “plane projection” the “planar projections?”
- Caption of Table 1: P are the planar moment of inertia instead of rotational angular moment. Please provide in table less digits (I think two digits after comma are enough).
- As to the fit: 1) what is the estimated error of the rotational transitions? 2) Values in parenthesis (looking also the .par file in the Supplementary materials) are not the standard errors. See: either (the text just before and after equation 4) https://www.sciencedirect.com/science/article/pii/S0022285216301801?via%3Dihub or the “ SPFIT parameter errors” at the PROSPE website (http://www.ifpan.edu.pl/~kisiel/asym/pickett/crib.htm#errors).
- I think is better to refer to estimated error on measurements with respect to linewidths (authors write to be about 150 kHz).
- In my opinion, the “Discussion” section is a bit confusing. In particular, among other aspects, I would suggest to add a picture to show higher energy conformers (to which the two reported barriers are related to), if I rightly understand.
- Please rephrase:” Our newly built cp-FTMW with a quadrature detection scheme was demonstrated with the capability to measure such non-volatile molecules in a supersonic jet.”
Author Response
Response to Reviewer comments
In particular, thanks to the use of a newly constructed broadband chirped-pulse Fourier transform microwave spectrometer, the extension of the rotational spectrum of benzophenone (already observed at higher frequency) in 2-8 GHz range has been performed, providing more accurate rotational parameters, also taking into account the previous data.
I think the paper can be suitable for publication in the journal but only after the authors have addressed these points:
- The authors write (lines 24-26):” Benzophenone might exist in galaxies like polycyclic aromatic hydrocarbons (PAHs) widespread through circumstellar envelopes and interstellar medium (ISM) of carbon rich stars [4,5]. The two reported references are not related to the astrochemical sentence the authors wrote. Please either correct them or remove the sentence.
Response 1:
Thank you for the revision advice. We noticed this misleading part and provided new references related to the IS detection of PAHs.
They are “
- Allamandola, L. J.; Tielens, A. G. G. M.; Barker, J. R. Interstellar Polycyclic Aromatic Hydrocarbons: The Infrared Emission Bands, the Excitation/Emission Mechanism, and the Astrophysical Implications. Astrophlys. J. Suppl. S. 1989, vol. 71, pp. 733.
- J. L.; Léger, A. A New Component of the Interstellar Matter: Small Grains and Large Aromatic Molecules. Puget, Annu. Rev. Astron Astr. 1989, Vol. 27, pp. 161–198.”
Also, ref 8 mentioned: “The global analysis of the microwave and millimeter wave data together resulted in the most precise values of the rotational constants to date and may be of interest to individuals attempting to identify these species (9-fluorenone
and benzophenone) via either radio astronomy or atmospheric remote sensing.” Although we could not find more evidence or references, we believe that benzophenone could potentially be a simple aromatic molecule linked to the formation of PAHs in the interstellar medium due to its photochemical reactivity.
- The authors write(lines 32-33):” Up to date, benzophenone has been investigated by microwave spectroscopy [5] and millimeter-wave spectroscopy [6].” In my opinion is better to write the frequency ranges since all the already measured and the present rotational transitions are within the microwave range.
Response 2:
Thank you for the revision advice. We noticed this misleading part and changed it to “Up to date, benzophenone has been investigated by microwave/millimeter-wave spectroscopy in the 8-14 GHz and 60–73 GHz region [8, 9].”
- Please re-write the sentence (lines 33-35) as something like:” This work aims to extend the laboratory measurements of the rotational spectrum of benzophenone at lower frequency…..”
Response 3:
Thank you for the revision advice. We noticed this misleading part and changed to “This work aims to extend the laboratory measurements of the rotational spectrum of benzophenone at lower frequency to further support the astrophysical observation and atmospheric remote sensing of benzophenone molecule.”
- Please rephrase (line 40): “The microwave spectrum of benzophenone in the range of 2-8 GHz…” in “The microwave spectrum of benzophenone in the 2-8 GHz range..”
Response 4:
Thank you for the revision advice. We noticed this misleading part and changed it to “The microwave spectrum of benzophenone in the 2 - 8 GHz range”
- I think it is better to rephrase the sentence:” The instrumentation of the spectrometer was based on the linear frequency sweeps (chirped pulses) generated by an arbitrary wave generator (AWG) achieved by the Pate and many other groups [9,10], but equipped with an economic quadrature detection plan.” as something like:” The spectrometer is based on the…. , as described by Pate and others [ ], “.
Response 5:
Thank you for the revision advice. We noticed this misleading part and changed it to “The spectrometer is based on the linear frequency sweeps (chirped pulses) generated by an arbitrary wave generator (AWG), as described by Pate and others [12, 13],”
- I think there is a typo in “Ciculator” in Figure 1.
- What is LO output?
Response 6:
Thank you for the revision advice. We noticed these mistakes and redraw the figure.
- Please rephrase the sentence:” A gas mixture of noble gas seeded with sample molecules was allowed to enter the vacuum chamber through a heating nozzle, i.e. a pulsed solenoid valve (Parker Series 9, 1.0 mm diameter) coupled with heating container (up to 250 ˚C) to obtain sufficient vapor pressure for seed chemicals.” as something like:” A gas mixture noble gas/sample has been obtained by flowing the….and expanded into the chamber…..”.
Response 7:
Thank you for the revision advice. We noticed this misleading part and changed it to “A sample gas mixture was obtained by flowing high purity argon (99.9% purity) through benzophenone right before the Parker nozzle, which is coupled with a band heater to obtain sufficient vapor pressure for solid chemicals.”
- Line 78/79: please correct the “lowest frequency range of benzophenone molecule” in “the 2-8 GHz range.
Response 8:
Thank you for the revision advice. We noticed this misleading part and changed it to “the 2-8 GHz range of benzophenone molecule”
- Line 80: do you mean “A total of 10 scans of 1 GHz range each”?
Response 9:
Yes. We noticed this misleading part and changed it to “A total of ten scans of 1 GHz range each”
- I think it is better to write in line 93:” Based on MP2, and DFT calculations, the isolated structure of benzophenone (not phenylacetone) was optimized….”. Please note that when you optimize to a minimum, what you obtain is the equilibrium structure (no ground vibrational state).
Response 10:
Thank you for the revision advice. We noticed this misleading part and changed it to “Based on ab initio [14] and density-functional theory (DFT) calculation [15, 16], the isolated structure of benzophenone was optimized by Gaussian09 software package [17].”
- Please correct (line 95): “the base set cc-pVDZ [15] was applied…” in “the cc-pVDZ basis set was employed…”.
Response 11:
Thank you for the revision advice. We noticed this misleading part and changed it to “the cc-pVDZ basis set was employed”
- Please rephrase the sentence (line 95-99) as something like:” Thanks to these calculations, the rotational parameters required for guiding the….”
Response 12:
Thank you for the revision advice. We noticed this misleading part and changed it to “Thanks to these calculations, the molecular constants or rotational constants required for guiding the initial spectroscopic assignment and fitting procedures.”
- In line 98: are they “planar projections”?
Response 13:
Thank you for the revision advice. We noticed this misleading part and changed it to planar projections. Finally, we redraw the figure.
- I think the authors can change the sentence at line 99 as (or something like): benzophenone using the optimized MP2/cc-pVDZ structure, are..”
Response 14:
Thank you for the revision advice. We noticed this misleading part and changed it to “benzophenone using the optimized MP2/cc-pVDZ structure, are”
- Line 101-103. I think is better to rephase as:” Table 1 shows the rotational parameters obtained at the MP2/cc-pVDZ and B3LYP/cc-pVDZ levels of theory as well as the moment of inertia and the planar moment of inertia of benzophenone…”
Response 15:
Thank you for the revision advice. We noticed this misleading part and changed it to “Table 1 shows the rotational parameters obtained at the MP2/cc-pVDZ and B3LYP/cc-pVDZ levels of theory as well as molecular parameters of benzophenone predicted by PMIFST (Principal Moments of Inertia from Structure) [19].”
- Please rephase the caption of Figure 5 (please don’t use opt, but explicit it). Are “plane projection” the “planar projections?”
Response 16:
Thank you for the revision advice. We noticed this misleading part and changed it to “optimized” and “planar projections”.
- Caption of Table 1: P are the planar moment of inertia instead of rotational angular moment. Please provide in table less digits (I think two digits after comma are enough).
Response 17:
Thank you for the revision advice. We noticed this misleading part and changed it to “the planar moment”. And we provided less digits in table.
- As to the fit: 1) what is the estimated error of the rotational transitions? 2) Values in parenthesis (looking also the .par file in the Supplementary materials) are not the standard errors. See: either (the text just before and after equation 4) https://www.sciencedirect.com/science/article/pii/S0022285216301801?via%3Dihub or the “SPFIT parameter errors” at the PROSPE website (http://www.ifpan.edu.pl/~kisiel/asym/pickett/crib.htm#errors).
- I think is better to refer to estimated error on measurements with respect to linewidths (authors write to be about 150 kHz).
Response 18:
Thank you for the revision advice. We noticed this misleading part and ran the piform program to update the standard deviation of rotational parameters and estimated error on measurements in Table 2.
- In my opinion, the “Discussion” section is a bit confusing. In particular, among other aspects, I would suggest to add a picture to show higher energy conformers (to which the two reported barriers are related to), if I rightly understand.
Response 19:
Thank you for the revision advice. We noticed this misleading part and changed it to
“Including this work, plenty of computational investigations have been performed on the benzophenone [8, 9, 23, 24]. In summary, ab initio and Density Functional Theory methods were mostly employed to optimize its structure,undoubtedly providing the paddle-wheel like C2 geometry for a global minimum with a single component of the dipole moment along the b principle axis. For the hindered internal rotation of the two phenyl groups, the latest MP2/cc-pVQZ structure from the van Wijngaarden group [8] by the second moment calculation determined a 32.9º torsional angle of the phenyl groups out of the a, b-principle plane in the gas-phase. In addition, two higher energy conformers regarding such hindered rotation with one T-like Cs symmetry and one C2v symmetry were predicted to stay 1084 cm-1 and 4267 cm-1 above the global minimum respectively, which excluded resolvable tunneling splitting associated with the internal rotations in the microwave spectrum and was confirmed by the free-jet experiments from our group and others [8, 9].”
Because the calculation of the hindered internal rotation was cited from ref. 8, a simple summarization without pictures might be appropriate.
- Please rephrase:” Our newly built cp-FTMW with a quadrature detection scheme was demonstrated with the capability to measure such non-volatile molecules in a supersonic jet.”
Response 20:
Thank you for the revision advice. We noticed this misleading part and changed it to “The newly built cp-FTMW spectrometer with a quadrature detection scheme and a heating nozzle demonstrated its capability to measure such non-volatile molecules in a supersonic expansion.”

Reviewer 2 Report
Attached.

Author Response
Response to Reviewer comments
The manuscript reports on the results of a rotational spectroscopic investigation on Benzophenone. Spectra were recorded in the 2 – 8 GHz region using a newly-built chirped-pulse Fourier transform microwave spectrometer. The spectroscopic work was supported by quantum chemical calculations for molecular structure determination. Authors have successfully recorded and assigned 138 new rotational transitions of benzophenone in its ground vibrational state. Accurate molecular parameters have been determined and compared with theoretically calculated and previously reported values. Improved rotational constants provide a useful guide for radioastronomical and atmospheric detection of Benzophenone.
From the scientific point of view, I judge this paper provide useful spectroscopic data, especially for the molecular spectroscopic community. Benzophenone is a relatively large molecule to handle using rotational spectroscopy and information on the new instrument design is interesting. These findings contribute to the growing body of spectroscopic knowledge about gas-phase molecular structures and rotational spectroscopic data.
In my opinion, therefore, the paper is appropriate for the Applied Sciences. My recommendation to the editor is “Minor Revision”. However, some part of the manuscript need be rewritten. In the following, I’m listing the comments I have for the authors.
Line 24: re-word the sentence. “galaxies like polycyclic aromatic hydrocarbons.” doesn’t sound right to me.
Response 1:
Thank you for the revision advice. We noticed this misleading part and changed it to
“Benzophenone might exist in galaxies similar to polycyclic aromatic hydrocarbons (PAHs), chemicals believed to be widespread through circumstellar envelopes and interstellar medium (ISM) of carbon rich stars [4, 5].”
Line 26: Instead of reference 4 and 5, I would provide new references that are more related to the IS detection of PAHs, review articles might be a better choice.
Response 2:
Thank you for the revision advice. We added new references related to the IS detection of PAHs.
They are “
- Allamandola, L. J.; Tielens, A. G. G. M.; Barker, J. R. Interstellar Polycyclic Aromatic Hydrocarbons: The Infrared Emission Bands, the Excitation/Emission Mechanism, and the Astrophysical Implications. Astrophlys. J. Suppl. S. 1989, vol. 71, pp. 733.
- J. L.; Léger, A. A New Component of the Interstellar Matter: Small Grains and Large Aromatic Molecules. Puget, Annu. Rev. Astron Astr. 1989, Vol. 27, pp. 161–198.”
Line 29: The sentence, “Under these conditions…” should be changed appropriately. Laboratory rotational spectroscopic data on benzophenone can be a useful guide for radioastronomical and atmospheric search of benzonitrile.
Response 3:
Thank you for the revision advice. We noticed this misleading part and changed it to “Under these conditions, laboratory rotational spectroscopic data on benzophenone can be a useful guide for radioastronomical and atmospheric search of this molecule.”
Line 45: Instead of reference 10, provide a review article that on chirped-pulse spectroscopy
Response 4:
Thank you for the revision advice. We noticed this misleading part and changed it to a new reference.
It is “12. Brown, G.G.; Dian, B.C.; Douglass, K.O.; Geyer, S.M.; Shipman, S.T.; Pate, B.H. A broadband Fourier transform microwave spectrometer based on chirped pulse excitation. Rev. Sci. Instrum. 2008, vol. 79, pp. 053103.”
Figure 1: Need to be improved significantly.
- Should be 10 MHz, not “10MHz”
- Giga samples per second => GS/s, not “Gs/s”
- Correct the direction of the arrow from microwave synthesizer to the mixer.
- Maybe it is more appropriate to name it as a “microwave synthesizer”? and correct the frequency range to 0.01 – 20 GHz?
- “Pulsed Nozzle” label: use arrows to point the nozzle.
Response 5:
Thank you for the revision advice. We noticed these mistakes and redrew the figure.
Line 50 – 77: This paragraph is missing some key information that readers would like to know. One of the highlights of this work is the custom-built spectrometer. As there are no previous literature on this spectrometer, I expect the authors to provide more details. Such as:
- Vacuum chamber – dimensions/diameter?
- Mirror – dimensions, materials used. Is it placed behind the nozzle? Is it attached to a flange?
- Nozzle – Is nozzle sitting inside the vacuum chamber?
- Fast-switch – model no?
- Mixers – model no?
- Any band pass filters in the circuit?
- Additional things that need be included in the paragraph: Rb-clock (is shown in the figure, but not mentioned in the text), down-conversion side of the circuit (is shown in the figure, but not effectively discussed in the text), what type of nozzle-heater is used (a band heater?), is there a voltage/temperature controller unit for the heater? What limits the maximum temperature to 250 OC? Delay generators? Sample information (purity, synthesized/purchased?)
- Are you using the same MW synthesizer for irradiation and down conversion? Explain more
I would include all this missing information and re-write the paragraph.
Response 6:
Thank you for the revision advice. We noticed all of the misleading parts and made the following changes: “The microwave spectrum of benzophenone in the 2 - 8 GHz range was collected by a newly developed broadband chirped-pulse Fourier transform microwave (cp-FTMW) spectrometer with a heating nozzle at Nanjing University of Science and Technology (NJUST) as shown in Figure 1, which has been described in detail previously [10, 11]. Basically, a vacuum chamber (1.0 m in length and 0.5 m in diameter) housing a reflection spherical aluminum mirror (350 mm in diameter and 800 mm in radius of curvature), a sample pulsed solenoid nozzle ((Parker Series 9, 1.0 mm in diameter)) and a feedhorn antenna, makes the main mechanical part of the spectrometer. The nozzle, penetrating the center of the mirror, is in a coaxial arrangement with the mirror, the antenna, and the chamber as well. The vacuum chamber was pumped by a molecular pump to maintain a background pressure around 1×10-5 Pa.
The spectrometer is based on the linear frequency sweeps (chirped pulses) generated by an arbitrary wave generator (AWG), as described by Pate and others [12, 13], but equipped with an economic quadrature detection plan. In this work, homodyne circuit design was applied to the cp-FTMW spectrometer system: the continuous microwave generated by the microwave source was split in two through a power distributor. Half of the microwave up-converted the broadband source generated by arbitrary waveform generator to excite the sample, and the other half down-converted the molecular excited signal to the oscilloscope acceptable bandwidth range. In this work, a 5 µs, 0–500 MHz chirped pulse from AWG (Siglent SDG6052x-e, 1.25 GS/s) was mixed in mixer 1 (Marki M1-0220LA Mixer) with the LO output from a microwave synthesizer (Anapico Apsyn420, 0.01 - 20 GHz) to generate an exciting pulse with 1 GHz chirp bandwidth, that was then amplified using a solid-state amplifier (BONN, 5W) before being coupled into a custom-built vacuum chamber containing one high gain horn antenna and a reflection spherical aluminum mirror for broadcasting and receiving the molecular emission signals. The molecular signal, i.e. the free induction decay (FID), was first amplified by a low-noise amplifier (Miteq AFS44 LNA, 1-18 GHz) before being split by a homemade quadrature detection structure to achieve image rejection. The image rejected FID signals were further amplified by two identical low-noise amplifiers ((Mini-Circuits ZFL 500LN+, 0.1-500 MHz), which were subsequently sampled by the two channels of a digitizer (Spectrum M4i.2223, 2.5 GS/s, 1.5 GHz bandwidth). Fast Fourier transformation of the resulting free induction decay leads to the broadband spectrum over the full bandwidth of the chirp. The whole sequence was repeated to produce multiple FIDs so that the resulting spectrum gives improved signal-to-noise ratio. Furthermore, DF4351 frequency synthesizer, the SDG6052x-e AWG and the M4i.2223 digitizer was referenced to a 10 MHz rubidium standard oscillator (SRS, FS725) for external stability. In order to avoid damage to the downstream amplifier and oscilloscope caused by high-power microwave pulses, a single-pole single-throw electronic switch (SPST, F9114A, 1-18 GHz) was set behind the mixer 2 (Marki M1-0220LA Mixer), which could be closed in time to protect the downstream electronic equipment.
Benzophenone (99.9% purity) was purchased from Aldrich without further purification. A sample gas mixture was obtained by flowing high purity argon (99.9% purity) through benzophenone right before the Parker nozzle, which is coupled with a band heater to obtain sufficient vapor pressure for solid chemicals. The temperature of the heater was regulated by a temperature controller to maintain a set temperature up to 250 ˚C.”
Line 71: 35 Pa vapor pressure for benzophenone at 25 OC? Can you provide a reference? Seems too low.
Response 7:
Thank you for the revision advice. We noticed this misleading part and delete the sentence. In this work, the heating temperature referred to previous work in ref. 8.
Line 80: Did you average chirp-up and chirp-down measurements for this 1 GHz? That need to be clearly mentioned in the text. Also, I assume your bandwidth is 0.5 GHz. Then how did you record 1 GHz at a time?
Response 8:
Thank you for the revision advice. We averaged chirp-up and chirp-down measurements up to 10,000 times and found the power spectra are stable compared to a single measurement. Also, in the text, we mentioned: “The data acquisition sequence was repeated and a total of 10,000 shots (230,000 FIDs) were averaged to produce the spectrum via Fast Fourier Transformation.”
For the bandwidth issue, although the AWG only generates a 0.5 GHz chirped pulse at a time, the 1 GHz bandwidth of the final exciting pulse is doubled by mixing the 0.5 GHz chirped pulse with a single frequency microwave (continuous wave). That single frequency is the LO for both chirp-up and chirp-down mixers, also the center of the FFT spectrum for one scan.
Line 93: “[11] and” the space between words is missing
Response 9:
Thank you for the revision advice. We noticed this mistake and added the space.
Line 94: correct “phenylacetone”
Response 10:
Thank you for the revision advice. We noticed this mistake and changed it to “benzophenone”.
Line 94: “was optimized by using Gaussian09”
Response 11:
Thank you for the revision advice. We noticed this misleading part and changed it to “was optimized by Gaussian09”.
Line 97: I would say molecular constants or rotational constants, instead of rotational parameters
Response 12:
Thank you for the revision advice. We noticed this misleading part and changed it to
“the molecular constants or rotational constants”
Line 99 and everywhere else it is used: correct to “MP2/cc-pVDZ”
Response 13:
Thank you for the revision advice. We noticed this mistake and changed it to “MP2/cc-pVDZ”.
Figure 5: Show the principal axes originating at the center of mass of the molecule. I would get rid of plane-projections and make the 3-D figure larger. Get rid of the atom-numbers, maybe use element symbols instead?
Response 14:
Thank you for the revision advice. We redraw the figure.
Line 104 & 110: What is PMIFST? Does it relate to Reference 16?
Response 15:
PMIFST is a software called “Principal Moments of Inertia from Structure”. It relates to the reference “19. Thompson, H.B. Calculation of Cartesian Coordinates and Their Derivatives from Internal Molecular Coordinates. J. Chem. Phys. 1967, vol. 47, pp. 3407–3410”.
Table 1: I would avoid Ia, Ib, Ic, Pa, Pb, Pc values, instead provide dipole moment information. Also, up to one decimal point data-precision is sufficient for rotational constants and dipole moment values.
Response 16:
Thank you for the revision advice. We noticed this misleading part and changed it to one decimal point data-precision for rotational constants and dipole moment values.
Line 111: I would get rid of this line completely. No need to describe them again as you have already mentioned them above the table.
Response 17:
Thank you for the revision advice. We noticed this misleading part and got rid of this line.
Line 115: Did you try using the S-reduction Hamiltonian? I am wondering whether that would make a difference. Also what is the kappa (asymmetric parameter) value? May be include that in the text.
Response 18:
Thank you for the revision advice. We noticed the mistake and the misleading part. We made the correction in the text: “The analysis of the pure rotational spectra in the microwave region was carried out using Watson A-reduced effective Hamiltonian [20]. ” We also added the kappa value in the Table 1.
Line 117 – 122: This is not new information and not a part of the results in this work. Just providing the reference is sufficient.
Response 19:
Thank you for the revision advice. We noticed this misleading part and changed it to several references.
Line 130 – Correct the eo – oe, dash-sign.
Response 20:
Thank you for the revision advice. We noticed this mistake and changed it “eo ↔ oe”
Table 2: Stick to the same type of parentheses in proving the units. Add a third column to compare the results in Reference 6 (table 6 of ref 6)
Response 21:
Thank you for the revision advice. We noticed this misleading part and changed it to
Table 2. Fit spectroscopic constants for benzophenone
|
Combined |
Previous work a |
|
|
Frequency range (GHz) |
2.0-14.0 |
8.0-14.0 |
60.0-73.0 |
|
A (MHz) |
1692.889219(31) |
1692.88929(23) |
1692.916(35) |
|
B (MHz) |
412.644660(11) |
412.644594(92) |
412.620(17) |
|
C (MHz) |
353.874564(11) |
353.874501(88) |
353.884(17) |
|
DJ (kHz) |
0.010635(19) |
0.01053(14) |
0.0122(17) |
|
DJK (kHz) |
-0.043444(83) |
-0.04372(54) |
-0.0517(60) |
|
DK (kHz) |
0.41417(59) |
0.4195(43) |
0.462(62) |
|
d1 (kHz) |
-0.0014406(23) |
-0.001442(17) |
|
|
d2 (kHz) |
-0.00099(15) |
-0.0003385(68) |
|
|
σ(kHz) |
2.6 |
8 |
110 |
|
N |
304 |
166 |
54 |
a Constants are from Ref. [8] and [9].
Line 145 – Reference is missing
Response 22:
Thank you for the revision advice. We noticed this misleading part and added the reference.
Line 158: correct to “optimize”
Response 23:
Thank you for the revision advice. We noticed this misleading part and changed it to “optimize”.
Line 165: correct to MP2/cc-pVQZ
Response 24:
Thank you for the revision advice. We noticed this mistake and changed it to “MP2/cc-pVDZ”.
Line 166: correct to B3LYP/cc-pVDZ
Response 25:
Thank you for the revision advice. We noticed this mistake and changed it to “B3LYP/cc-pVDZ”.
Line 235: Correct the format of the reference
Response 26:
Thank you for the revision advice. We noticed this mistake and changed it to “18. Jr., T.H.D. Gaussian basis sets for use in correlated molecular calculations. I. The atoms boron through neon and hydrogen. J. Chem. Phys. 1989, vol. 90, pp. 1007.”
Lines 166 – 172: Authors should put more effort to provide a thorough summary of important molecular structural information based on the Ref 5 and Ref 6. Elaborate the content in Lines 166 – 172. Additionally, authors can mention about the Global Analysis and hence improved rotational constants given in the Ref-5.
Response 27:
Thank you for the revision advice. We added more content regarding the two references in the summary:
“Hamiltonian employed here has accurately described the observed spectral features in the microwave range. But for certain centrifugal distortion constants, such as d2, a global fitting from higher microwave frequency data [8, 9] can definitely provide more accurate results due to high Ka and Kc transitions available.
Including this work, plenty of computational investigations have been performed on the benzophenone [8, 9, 23, 24]. In summary, ab initio and Density Functional Theory methods were mostly employed to optimize its structure,undoubtedly providing the paddle-wheel like C2 geometry for a global minimum with a single component of the dipole moment along the b principle axis. For the hindered internal rotation of the two phenyl groups, the latest MP2/cc-pVQZ structure from the van Wijngaarden group [8] by the second moment calculation determined a 32.9º torsional angle of the phenyl groups out of the a, b-principle plane in the gas-phase. In addition, two higher energy conformers regarding such hindered rotation with one T-like Cs symmetry and one C2v symmetry were predicted to stay 1084 cm-1 and 4267 cm-1 above the global minimum respectively, which excluded resolvable tunneling splitting associated with the internal rotations in the microwave spectrum and was confirmed by the free-jet experiments from our group and others [8, 9].”
Supporting Information
SPFIT results are not necessary
Table S1 – “MHz” are in a smaller font. That need to be corrected. Use equation insert to properly show quantum number labels, such as ??′ not Ka’
Additionally, provide x, y, z coordinates of the optimized structures using both methods in two new tables.
Response 28:
Thank you for the revision advice. We corrected these mistakes and provided x, y, z coordinates of optimized structures using both methods in two new tables.

Round 2
Reviewer 1 Report
My compliments to the authors.
I would just suggest to the authors to be consistent in the fit: whether it is in A-reduction, please either use the corrected symbols for the quartic distortion constants otherwise change the A-reduction in the main text to S-reduction.
Author Response
Thank you for the revision advice. We noticed this misleading part and changed it to "The analysis of the pure rotational spectra in the microwave region was carried out using Watson S-reduced effective Hamiltonian. "